# The Accuracy Assessment of the PREM and AK135-F Radial Density Models

**DOI:** 10.3390/s22114180

**Published:** 2022-05-31

**Authors:** Robert Tenzer, Yuting Ji, Wenjin Chen

**Affiliations:** 1Department of Land Surveying and Geo-Informatics, Hong Kong Polytechnic University, Hong Kong; robert.tenzer@polyu.edu.hk (R.T.); jiyuting18@mails.ucas.ac.cn (Y.J.); 2College of Earth and Planetary Sciences, University of Chinese Academy of Sciences, Beijing 101408, China; 3School of Civil and Surveying & Mapping Engineering, Jiangxi University of Science and Technology, Ganzhou 341000, China

**Keywords:** earth synthetic model, PREM, AK135-F density, geoidal geopotential value, gravity

## Abstract

The Earth’s synthetic density and gravitational models can be used to validate numerical methods for global (or large-scale) gravimetric forward and inverse modelling formulated either in the spatial or spectral domains. The Preliminary Reference Earth Model (PREM) density parameters can be adopted as a 1-D reference density model and further refined using more detailed 2-D or 3-D crust and mantle density models. Alternatively, the AK135-F density parameters can be used for this purpose. In this study, we investigate options for a refinement of the Earth’s synthetic density model by assessing the accuracy of available 1-D density models, specifically the PREM and AK135-F radial density parameters. First, we use density parameters from both models to estimate the Earth’s total mass and compare these estimates with published results. We then estimate the Earth’s gravity field parameters, particularly the geoidal geopotential number W_0_ and the mean gravitational attraction and compare them with published values. According to our results, the Earth’s total mass from the two models (the PREM and the AK135-F) differ less than 0.02% and 0.01%, respectively, when compared with the value adopted by the International Astronomical Union (IAU). The geoidal geopotential values of the two models differ from the value adopted by the IAU by less than 0.1% and 0.04%, respectively. The values of the mean gravitational attraction of the two models differ less than 0.02% and 0.08%, respectively, when compared with the value obtained from the geocentric gravitational constant and the Earth’s mean radius. These numerical findings ascertain that the PREM and AK135-F density parameters are suitable for defining a 1-D reference density model.

## 1. Introduction

A number of seismic velocities and mass density models have been developed based on the analysis of tomographic data while incorporating geophysical constraints. Dziewonski prepared the Parametric Earth Model (PEM) [1]. This model is defined by piece-wise continuous analytical functions of the radial density and velocity variations provided individually for the oceanic (PEM-O) and continental (PEM-C) lithosphere down to the depth of 420 km, while below this depth, these two models are identical. This model also comprises an averaged function for the whole lithosphere (PEM-A). Later, Dziewonski and Anderson compiled the Preliminary Reference Earth Model (PREM) [2]. This model provides information on elastic properties, the anelastic attenuation factor, density, pressure and gravity within the Earth’s interior. The parameterized velocity model IASP91 prepared by [3] provides travel-time characteristics of the main seismic phases. Kennett compiled the AK135-F model by augmenting the AK135 seismic velocity model with the model of density and seismic quality factors prepared by Montagner and Kennett [4,5]. van der Lee and Nolet updated the PEM-C continental lithospheric parameters by replacing the high and low-velocity zones in the PEM-C with the constant S-wave velocity of 4.5 km s^−1^ within the uppermost mantle down to the depth of 210 km [6]. Kustowski derived the transversely isotropic reference model STW105 of the Earth’s interior [7]. Simmons developed the GyPSuM tomographic model of the mantle’s (P and S) seismic velocities and density through a simultaneous inversion of seismic body-wave travel times and geodynamic observables, including the free-air gravity anomalies, the tectonic plate divergence, the dynamic surface topography and the excess ellipticity of the core–mantle boundary [8]. They also incorporated mineral physics constraints in order to link seismic velocities and wave speeds with an underlying hypothesis that temperature is a principal cause of heterogeneities in the non-cratonic mantle. For a comprehensive summary of global seismic models, we refer readers to [9].

Seismic velocities models are often used as the basis for seismic tomography, while the applicability of density models is typically restricted to only lithospheric density models. In geodetic applications, for instance, topographic density models are used for gravimetric geoid modelling. The topographic density information (that also incorporates digital terrain models), together with an adopted hypothesis about a compensation mechanism, is used to compile isostatic gravity maps [10]. In geophysical applications, the crustal and lithospheric density models are used to compile the Bouguer and mantle gravity maps [11,12,13,14,15].

The 1-D reference density models mentioned in the first paragraph could be refined by incorporating 2-D or 3-D global lithospheric and mantle density models. Whereas reliable 3-D mantle density models are rare, a number of 3-D crustal and lithospheric density models have been published. Nataf and Ricard derived the crustal and upper-mantle density model based on the analysis of seismic data and additional constraints, such as the heat flow and chemical composition [16]. Mooney compiled the CRUST5.0 global crustal model with a 5 × 5° spatial resolution [17]. Later, the updated global crustal model CRUST2.0 was compiled with a 2 × 2° resolution by [18]. Both models were prepared from seismic data published before 1995 and by using more detailed information about the ice and sediment thickness. CRUST1.0 is the most recent version, complied globally with a 1 × 1° resolution [19]. CRUST1.0 consists of the ice, seawater, (upper, middle and lower) sediments and (upper, middle and lower) consolidated (crystalline) crustal layers. In addition, the lateral density structure of the uppermost mantle was incorporated in CRUST2.0 and CRUST1.0. Pasyanos compiled the LITHO1.0 global seismic model of the lithosphere, including the asthenosphere [20]. This model was prepared in order to fit the high-resolution (Love and Rayleigh) surface wave dispersion maps using the CRUST1.0 crust data and the LLNL-G3D upper mantle model as the a priori information [21]. Compared to similar 3-D density and velocity models, this model also provides information about the lithosphere–asthenosphere boundary. Hirt and Rexer constructed the Earth2014 global model that provides information about topographic heights, ocean-floor depths and the polar glacier bedrock relief [22]. Chen and Tenzer compiled the Earth’s Spectral Crustal Model 180 (ESCM180) by augmenting the Earth2014 and CRUST1.0 models [23].

As afore stated, the lithospheric and mantle density models could be used to refine 1-D density models (such as the PREM or AK135-F) in order to provide a more realistic representation of the Earth’s inner density structure. Since many parts of the world are not yet sufficiently covered by tomographic surveys, such refinement is not simple. Moreover, the direct relationship between seismic velocities and mass density values does not exist, as the density distribution depends on many other factors, such as temperature, mineral composition and pressure. In spite of these practical and theoretical limitations, synthetic density and gravitational models can be used to assess the numerical performance of gravimetric forward and inverse modelling methods. Synthetic density models have been used for the testing of numerical procedures involved in gravimetric geoid modelling [24,25,26]. Other examples of possible applications could be given in studies of the sediment bedrock morphology [27], the lithospheric and mantle density structure [28,29], the Moho geometry [15,28,29], the dynamic and residual topography [30,31,32] or the oceanic lithosphere thermal contraction and its isostatic rebalance [33].

To inspect the possibilities of refining synthetic density models, we evaluated the accuracy and resolution of existing 1-D density models, particularly the PREM and AK135-F density parameters. The assessments were carried out directly and indirectly. In the former, we compared the Earth’s total mass computed from the PREM and AK135-F density parameters with the published results obtained from an orbital analysis of satellites. In the latter, we computed the geoidal geopotential value W_0_ and the mean gravitational attraction from the PREM and AK135-F density parameters and compared them with the published estimates from the analysis of satellite altimetry and gravitational data. A brief description of the PREM and AK135-F models (in Section 2) is followed by a summary of numerical procedures used to validate these models (in Section 3). The results are presented in Section 4, and the study is concluded in Section 5.

## 2. Reference Density Models

The PREM structure was designed to fit various geophysical observations, including the travel times of the body-wave phase, the surface-wave dispersion and the free-oscillation centre frequency measurements while taking into consideration the total mass and volume of Earth and its rotational inertia. The PREM parameters of P and S wave velocities, density, pressure and the shear and bulk seismic quality factors are defined by piece-wise functions for spherically-homogenous stratigraphic layers, specifically the inner and outer core, the core–mantle boundary zone, the lower mantle, the inner (two layers) and outer transition zone, the low-velocity zone, the lithospheric mantle, the inner and outer crust as well as the ocean. The concept of weighted average was adopted for the first 100 km of depth, assuming that the oceanic crust covers two-thirds of the Earth’s surface and that the average Moho depth under the oceanic crust is 11 km and 35 km under the continental crust, yielding the global average for the whole Earth of 19 km in the Moho depth. The first 100-km-deep layer of the lithosphere was then divided into the 5 km layer of the ocean, 12 km layer of the upper crust with a density of 2600 kg m^−3^, the 9.4 km layer of the lower crust of density 2900 kg m^−3^ and the 55.6 km thick layer of the low-velocity zone. The PREM is transversely isotropic (i.e., spherically symmetric anisotropy model for which the two shear-wave components travel at different speeds) at depths between 80 and 220 km in the upper mantle to simultaneously fit the Love and Rayleigh-wave measurements. The PREM density parameters are summarized in Table 1.

The AK135 velocity model was augmented with the density and seismic quality factor models by combining the study of travel times with those of free oscillations. This velocity and density model incorporates the velocity model at depths below 120 km prepared by [4] and the modified density and the shear and bulk seismic quality factors compiled by [5]. The AK135-F parameters of the density, the (P and S) wave velocities and the shear and bulk seismic quality factors were defined by discrete values for numerous spherically symmetric layers, where these values are either constant or provided by two different values for the upper and lower bound of a particular spherical layer (see Table 2). The PREM and AK135-F models are quite similar, except for the discontinuity in the PREM at a depth of 22 km that is absent in the AK135-F.

## 3. Method

The PREM radially varying density distribution (for individual layers) is described by the following polynomial function
(1)$$\rho =\rho_{0}+\sum_{j=1}^{J}\rho_{i}{\left(\frac{r}{R}\right)}^{j},$$
where different orders (up to 3) of the upper summation index J are used depending on a particular layer. The transition zone, for instance, is defined by density parameters up to the third order, while the ocean layer is only a constant density (cf. Table 1).

The mass of the PREM spherically symmetric volumetric layers was computed as
(2)$$M=\rho_{0}\iint_{\Phi }\int_{r^{\prime }=r_{L}\left(\Omega^{\prime }\right)}^{r_{U}\left(\Omega^{\prime }\right)}{r^{\prime }}^{2}dr^{\prime }d\Omega^{\prime }+\sum_{j=1}^{J}\rho_{i}\iint_{\Phi }\int_{r^{\prime }=r_{L}\left(\Omega^{\prime }\right)}^{r_{U}\left(\Omega^{\prime }\right)}\frac{{r^{\prime }}^{j+2}}{R^{j}}dr^{\prime }d\Omega^{\prime },$$
where R is the Earth’s mean radius, and $r_{U}$ and $r_{L}$ are radii of the upper and lower bounds of the volumetric mass layer, respectively.

The gravitational potential V of individual layers was computed using the following equation
(3)$$\begin{array}{l} V={G\rho }_{0}\iint_{\Phi }\int_{r^{\prime }=r_{L}\left(\Omega^{\prime }\right)}^{r_{U}\left(\Omega^{\prime }\right)}{r^{\prime }}^{2}\;\ell^{-1}\left(R,\psi ,r^{\prime }\right)\;dr^{\prime }d\Omega^{\prime }\\ +\;G\sum_{j=1}^{J}\rho_{i}\iint_{\Phi }\int_{r^{\prime }=r_{L}\left(\Omega^{\prime }\right)}^{r_{U}\left(\Omega^{\prime }\right)}\frac{{r^{\prime }}^{j+2}}{R^{j}}\;\ell^{-1}\left(R,\psi ,r^{\prime }\right)\;dr^{\prime }d\Omega^{\prime } \end{array}$$
where G = 6.67 × 10^−^^11^ m^3^ kg^−1^ s^−2^ is the Newton’s gravitational constant, ℓ is the Euclidean spatial distance between positions of computation point $\left(R,\Omega \right)$ and integration point $\left(r^{\prime },\Omega^{\prime }\right)$ and ψ is their respective spherical distance.

The gravitational attraction g≅−∂V/∂r was computed as follows
(4)$$\begin{array}{c} g\left(R,\Omega \right)=-G\rho_{0}\iint_{\Phi }\int_{r^{\prime }=r_{L}\left(\Omega^{\prime }\right)}^{r_{U}\left(\Omega^{\prime }\right)}{\left.\frac{\partial \ell^{-1}\left(r,\psi ,r^{\prime }\right)}{\partial r}\right|}_{r=R}\;{r^{\prime }}^{2}dr^{\prime }d\Omega^{\prime }\\ -G\sum_{j=1}^{J}\rho_{i}\iint_{\Phi }\int_{r^{\prime }=r_{L}\left(\Omega^{\prime }\right)}^{r_{U}\left(\Omega^{\prime }\right)}\frac{{r^{\prime }}^{j+2}}{R^{j}}\;{\left.\frac{\partial \ell^{-1}\left(r,\psi ,r^{\prime }\right)}{\partial r}\right|}_{r=R}\;dr^{\prime }d\Omega^{\prime }, \end{array}$$

The gravitational potential of a homogenous spherical shell of a uniform density between the lower and upper bounds $r_{L}$ and $r_{U}$ ($r_{L}<\;r_{U}$) in the first-term on the right-hand side of Equation (3) was computed using the following expression [34]
(5)$$\begin{array}{l} V\left(R,\Omega \right)=G\;\rho_{0}\;\iint_{\Phi }\int_{\;r^{\prime }=r_{L}}^{\;r_{U}}\ell^{-1}\left(R,\psi ,\;r^{\prime }\right)\;{r^{\prime }}^{2}\;dr^{\prime }d\Omega^{\prime }\\ \;\;\;\;=4\pi \;G\;\rho_{0}\;\frac{1}{R}\left[r_{L}^{2}\left(r_{U}-r_{L}\right)+r_{L}{\left(r_{U}-r_{L}\right)}^{2}+\frac{1}{3}{\left(r_{U}-r_{L}\right)}^{3}\right]\left(R\geq r_{U}\right), \end{array}$$

From Equation (5), the expression for the gravitational potential of a homogenous sphere of a uniform density $\rho_{0}$ and a radius $r_{U}$ reads
(6)$$\begin{array}{c} V\left(R,\Omega \right)=G\;\rho_{0}\;\iint_{\Phi }\int_{\;r^{\prime }=0}^{\;r_{U}}\ell^{-1}\left(R,\psi ,\;r^{\prime }\right)\;{r^{\prime }}^{2}\;dr^{\prime }d\Omega^{\prime }\\ =\frac{4}{3}\pi G\;\rho_{0}\;\frac{r_{U}^{3}}{R}\left(R\geq r_{U}\right), \end{array}$$

The gravitational attraction of a homogenous spherical shell of a uniform density between the lower and upper bounds $r_{L}$ and $r_{U}$ ($r_{L}<r_{U}$) in the first-term on the right-hand side of Equation (4) was computed as follows [34]
(7)$$\begin{array}{l} \;\;g\left(R,\Omega \right)=-G\;\rho_{0}\;\iint_{\Phi }\int_{\;r^{\prime }=r_{L}}^{\;r_{U}}{\left.\frac{\partial \ell^{-1}\left(r,\psi ,r^{\prime }\right)}{\partial r}\right|}_{r=R}\;{r^{\prime }}^{2}\;dr^{\prime }d\Omega^{\prime }\\ \;\;\;\;\;=-4\pi G\rho_{0}\frac{1}{R^{2}}\left[r_{L}^{2}\left(r_{U}-r_{L}\right)+r_{L}{\left(r_{U}-r_{L}\right)}^{2}+\frac{1}{3}{\left(r_{U}-r_{L}\right)}^{3}\right]\\ \left(R\geq r_{U}\right), \end{array}$$

Consequently, the expression for the gravitational attraction of a homogenous sphere of a uniform density $\rho_{0}$ and a radius $r_{U}$ is given by
(8)$$\begin{array}{c} g\left(R,\Omega \right)=G\;\rho_{0}\;\iint_{\Phi }\int_{\;r^{\prime }=0}^{\;r_{U}}{\left.\frac{\partial \ell^{-1}\left(r,\psi ,r^{\prime }\right)}{\partial r}\right|}_{r=R}\;{r^{\prime }}^{2}\;dr^{\prime }d\Omega^{\prime }\\ =\frac{4}{3}\pi \;G\;\rho_{0}\;\frac{r_{U}^{3}}{R^{2}}\left(r\geq r_{U}\right), \end{array}$$

We note that the expressions in Equations (5) and (7) were used to compute the gravitational potential and attraction of the PREM zero-order density terms, except for the inner-zone, which was computed by using the expressions in Equations (6) and (8). The elliptical integrals in the second term on the right-hand side of Equations (3) and (4) were solved numerically by applying the Gaussian quadrature rule. Since most of the AK135-F density layers are described either by a constant density value or by two different density values defined individually for the upper and lower bound (cf. Table 2), we first converted the AK135-F density description into that used for the PREM, while using only zero and first-order density terms (due to the fact that only linear density changes are considered within individual AK135-F layers). The Earth’s total mass was computed as a sum of individual contributions of the PREM or AK135-F spherical density layers. Similarly, the geoidal geopotential value and the mean gravitational attraction were obtained as the sum of individual contributions. The results are presented and compared with published values next.

## 4. Results

The Earth’s total mass M_Earth_, the geoidal geopotential value W_0_ and the mean gravitational attraction g estimated from the PREM and AK135-F density parameters are summarized in Table 3 and Table 4, respectively. The tables also provide the mass, gravitational potential and attraction of individual layers. Since the AK135-F density parameters are provided for more than 140 individual layers, we presented the results similarly to that used for the PREM components of the Earth’s interior. We note that there are some inconsistencies because the AK135-F depth structure does not coincide exactly with the PREM depth layers.

The Earth’s parameters M_Earth_, W_0_ and g estimated from the PREM and AK135-F density parameters were compared with the corresponding values adopted in the Earth’s science applications. In addition, we made comparisons with the GRS80 reference ellipsoid parameters [35].

As seen in Table 3, the Earth’s total mass estimated by using the PREM density parameters very closely agrees with the value of 5.9722 ± 0.0006 × 10^24^ kg adopted by the International Astronomical Union (IAU). Even better agreement was achieved for the Earth’s total mass obtained from the AK135-F density parameters (Table 4).

The geoidal geopotential values estimated from the PREM and AK135-F density parameters very closely agree with the conventional value of 62,636,856.0 ± 0.5 m^2^ s^−2^ adopted by the IAU, as well as by the International Earth Rotation and Reference System Service’s [36,37], while the AK135-F provides a better fix when compared with the PREM.

Despite the parameter W_0_ estimated from the PREM density parameters differing less than 0.1% from the corresponding value adopted by the IAU, the application of this value in the geoid modelling will introduce a large offset. When taking into consideration the normal gravitational potential U_0_ = 62,636,860.850 m^2^ s^−2^ of the GRS80 reference ellipsoid, the difference between the gravitational potential on the geoid and the normal potential on the reference ellipsoid (i.e., W_0_ − U_0_) exceeds 61,320 m^2^ s^−2^ (in absolute value). In terms of the geoid modelling (i.e., (W_0_ − U_0_)/γ_0_, where γ_0_ denotes the normal gravity at the reference ellipsoid), such value corresponds to the geoid error of roughly 6 km, while the global geoidal undulations are mostly within ±0.1 km. Even if taking into consideration the parameter W_0_ estimated from the AK135-F density parameters, the geoid error would exceed 2 km.

Finally, we compared our estimates of the geoidal geopotential and the mean gravitational attraction with the gravitational potential and attraction computed using the geocentric gravitational constant (GM = 3,986,005 × 10^8^ m^3^ s^−2^) and the Earth’s mean radius (R = 6371 × 10^3^ m). The results showed that differences in terms of the gravitational potential are roughly 0.02% (for the PREM) and 0.08% (for the AK135-F). The same relative differences were found for the gravitational attraction. We note that the comparison of the mean gravitational potential from the PREM and AK135-F density parameters with the normal gravity of the GRS reference ellipsoid is not optimal as this value changes with the geodetic latitude, while the geoidal geopotential is a constant value.

## 5. Concluding Remarks

We have used the PREM and AK135-F density parameters defined by means of depth-density changes within individual layers and described in terms of the piece-wise Roche’s model [38] to estimate the Earth’s total mass, the geoidal geopotential value and the mean gravitational attraction. We then compared these values with the published values and numerical results.

The comparison revealed that our estimates closely agree with adopted values of the Earth’s total mass and the geoidal geopotential value. Particularly good agreement was found for the estimates based on using the AK135-F density parameters with relative differences of less than 0.01% for the total mass of Earth and of less than 0.04% for the geoidal geopotential value, both parameters adopted by the IAU.

Interestingly, the PREM density parameters provide a better fit with the gravitational potential and attraction (both computed as a function of the geocentric gravitational constant GM and the Earth’s mean radius R) than the AK135-F model, with relative differences of 0.02% (for the PREM) and 0.08% (for the AK135-F).

From these numerical findings, we concluded that the PREM and AK135-F density parameters very closely agree with the Earth’s parameters. Consequently, both models can be adopted for a 1-D reference density parameterization applied for the preparation of the Earth’s synthetic density and gravitational models. Nevertheless, careful considerations have to be made in the context of using these density parameters for some particular applications. We provided such examples in the geoid modelling, where a large error (bias) is introduced when using the PREM and AK135-F density parameters. Nevertheless, a simple correction can be applied to remove the systematic bias between W_0_ and U_0_.

## Figures and Tables

**Table 1 sensors-22-04180-t001:** The PREM density parameters according to [2].

Layer	Radius (km)		ρ_0_ (kg m^−3^)	ρ_1_ (kg m^−3^)	ρ_2_ (kg m^−3^)	ρ_3_ (kg m^−3^)
Inner core		0–1221.5	13,088.5	-	−8838.1	-
Outer core		1221.5–3480.0	12,581.5	−1263.8	−3642.6	−5528.1
Lower mantle		3480.0–5701.0	7.9565	−6.4761	5.5283	−3.0807
Transition zone	Zone 1	5701.0–5771.0	5.3197	−1.4836	-	-
	Zone 2	5771.0–5971.0	11.2494	−8.0298	-	-
	Zone 3	5971.0–6151.0	7.1089	−3.8045	-	-
LVZ		6151.0–6291.0	2691.0	0.6924	-	-
LID		6291.0–6346.6	2691.0	0.6924	-	-
Crust	Zone 1	6346.6–6356.0	2900	-	-	-
	Zone 2	6356.0–6368	2600	-	-	-
Ocean		6368.0–6371.0	1020	-	-	-

**Table 2 sensors-22-04180-t002:** The AK135-F density parameters according to [4].

Depth (km)	Density (g cm^−3^)	Depth (km)	Density (g cm^−3^)	Depth (km)	Density (g cm^−3^)	Depth (km)	Density (g cm^−3^)
0–3	1.02	1255	4.7266	2939.33	9.9942	4751.25	11.9098
3–3.3	2	1304.5	4.7528	2989.66	10.0722	4801.58	11.9414
3.3–10	2.6	1354	4.779	3039.99	10.1485	4851.91	11.9722
10	2.92	1403.5	4.805	3090.32	10.2233	4902.24	12.0001
18	2.92	1453	4.8307	3140.66	10.2964	4952.58	12.0311
18	3.641	1502.5	4.8562	3190.99	10.3679	5002.91	12.0593
43	3.5801	1552	4.8817	3241.32	10.4378	5053.24	12.0867
80	3.502	1601.5	4.9069	3291.65	10.5062	5103.57	12.1133
80	3.502	1651	4.9321	3341.98	10.5731	5153.5	12.1391
120	3.4268	1700.5	4.957	3392.31	10.6385	5153.5	12.7037
120	3.4268	1750	4.9817	3442.64	10.7023	5204.61	12.7289
165	3.3711	1799.5	5.0062	3492.97	10.7647	5255.32	12.753
210	3.3243	1849	5.0306	3543.3	10.8257	5306.04	12.776
210	3.3243	1898.5	5.0548	3593.64	10.8852	5356.75	12.798
260	3.3663	1948	5.0789	3643.97	10.9434	5407.46	12.8188
310	3.411	1997.5	5.1027	3694.3	11.0001	5458.17	12.8387
360	3.4577	2047	5.1264	3744.63	11.0555	5508.89	12.8574
410	3.5068	2096.5	5.1499	3794.96	11.1095	5559.6	12.8751
410	3.9317	2146	5.1732	3845.29	11.1623	5610.31	12.8917
460	3.9273	2195.5	5.1963	3895.62	11.2137	5661.02	12.9072
510	3.9233	2245	5.2192	3945.95	11.2639	5711.74	12.9217
560	3.9218	2294.5	5.242	3996.28	11.3127	5762.45	12.9351
610	3.9206	2344	5.2646	4046.62	11.3604	5813.16	12.9474
660	3.9201	2393.5	5.287	4096.95	11.4069	5863.87	12.9586
660	4.2387	2443	5.3092	4147.28	11.4521	5914.59	12.9688
710	4.2986	2492.5	5.3313	4197.61	11.4962	5965.3	12.9779
760	4.3565	2542	5.3531	4247.94	11.5391	6016.01	12.9859
809.5	4.4118	2591.5	5.3748	4298.27	11.5809	6066.72	12.9929
859	4.465	2640	5.3962	4348.6	11.6216	6117.44	12.9988
908.5	4.5162	2690	5.4176	4398.93	11.6612	6168.15	13.0036
958	4.5654	2740	5.4387	4449.26	11.6998	6218.86	13.0074
1007.5	4.5926	2740	5.6934	4499.6	11.7373	6269.57	13.01
1057	4.6198	2789.67	5.7196	4549.93	11.7737	6320.29	13.0117
1106.5	4.6467	2839.33	5.7458	4600.26	11.8092	6371	13.0122
1156	4.6735	2891.5	5.7721	4650.59	11.8437	-	-
1205.5	4.7001	2891.5	9.9145	4700.92	11.8772	-	-

**Table 3 sensors-22-04180-t003:** Mass, gravitational potential and attraction of the PREM layers.

Layer		Potential (m^2^ s^−2^)	Attraction (m s^−2^)	Mass (kg)	Radius (km)
Inner core		1.0312 × 10^6^	0.1619	9.8433 × 10^22^	0–1221.5
Outer core		1.9288 × 10^7^	3.0274	1.8411 × 10^24^	1221.5–3480.0
Lower mantle		3.0802 × 10^7^	4.8347	2.9402 × 10^24^	3480.0–5701.0
Transition zone	Zone 1	1.2079 × 10^6^	0.1896	1.1530 × 10^23^	5701.0–5771.0
	Zone 2	3.4928 × 10^6^	0.5482	3.3341 × 10^23^	5771.0–5971.0
	Zone 3	3.0374 × 10^6^	0.4768	2.8994 × 10^23^	5971.0–6151.0
LVZ		2.4018 × 10^6^	0.3770	2.2927 × 10^23^	6151.0–6291.0
LID		9.8714 × 10^5^	0.1549	9.4229 × 10^22^	6291.0–6346.6
Crust	Zone 1	1.4476 × 10^5^	0.0227	1.3819 × 10^22^	6346.6–6356.0
	Zone 2	1.6625 × 10^5^	0.0261	1.5869 × 10^22^	6356.0–6368
Ocean		1.6343 × 10^4^	0.0026	1.5601 ×10^21^	6368.0–6371.0
PREM		W_0_ = 62,575,540	g = 9.8219	M_Earth_ = 5.9732 × 10^24^	R = 6371

**Table 4 sensors-22-04180-t004:** Mass, gravitational potential and attraction of the AK135-F model provided in the form of the PREM layers.

Layer		Potential (m^2^ s^−2^)	Attraction (m s^−2^)	Mass (kg)	Radius (km)
Inner core		1.0153 × 10^6^	0.1594	9.6916 × 10^22^	0–1217.5
Outer core		1.9285 × 10^7^	3.0270	1.8409 × 10^24^	1217.5–3479.5
Lower mantle	Zone 1	1.4458 × 10^6^	0.2269	1.3801× 10^23^	3479.5–3631.0
	Zone 2	2.9523 × 10^7^	4.6340	2.8181 × 10^24^	3631.0–5711.0
Transition zone *	Zone 1	4.3979 × 10^6^	0.6903	4.1981 × 10^23^	5711.0–5961.0
	Zone 2	3.3201 × 10^6^	0.5211	3.1692 × 10^23^	5961.0–6161.0
LVZ		2.2382 × 10^6^	0.3513	2.1365× 10^23^	6161.0–6291.0
LID		1.1529 × 10^6^	0.1810	1.1006 × 10^23^	6291.0–6353.0
Crust	Zone 1	1.2428 × 10^5^	0.0195	1.1863 × 10^22^	6353.0–6361.0
	Zone 2	9.2889 × 10^4^	0.0146	8.8668 × 10^21^	6361.0–6367.7
	Zone 3	3.2029 × 10^3^	0.0005	3.0574 × 10^20^	6367.7–6368.0
Ocean		1.6343 × 10^4^	0.0026	1.5601 × 10^21^	6368.0–6371.0
AK135-F		W_0_ = 62,615,208	g = 9.8282	M_Earth_ = 5.9770 × 10^24^	R = 6371

* Note that only two layers are defined for the transition zone, while two were used for the lower mantle (instead of one in the PREM), and the crust was divided into three layers (instead of two in the PREM).

## Data Availability

Not applicable.

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
