# Peer review of "The Accuracy Assessment of the PREM and AK135-F Radial Density Models"

_sensors, 2022, doi:10.3390/s22114180_

Round 1

Reviewer 1 Report

The paper deals with assessment of the PREM and AK135-F radial density models by gravity forward modeling. It is very well written and clear. Here by I recommend it to be published with the following minor comments that should be addressed: For the abbreviation PREM should be have full name when it first appeal. In line 148-149, the sentence The PREM radially-varying density distribution (for individual layers) is described by the following function should be rewritten as The PREM radially-varying density distribution (for individual layers) is described by the following polynomial function In line 154, the sentence where m is the Earths mean radius should be rewritten as where R is the Earths mean radius’ In equation (3), the left hand of the equation should be not In line 158, they lost symbol G of the Newtons gravitational constant

Author Response

Annotated respond

We thank Editors for their constructive comments. We addressed them below.

The paper deals with assessment of the PREM and AK135-F radial density models by gravity forward modeling. It is very well written and clear. Here by I recommend it to be published with the following minor comments that should be addressed:

For the abbreviation PREM should be have full name when it first appeal.

Corrected

In line 148-149, the sentence The PREM radially-varying density distribution (for individual layers) is described by the following function should be rewritten as The PREM radially-varying density distribution (for individual layers) is described by the following polynomial function

Corrected

In line 154, the sentence where m is the Earths mean radius should be rewritten as where R is the Earths mean radius

Changed

In equation (3), the left hand of the equation should be not In line 158, they lost symbol G of the Newtons gravitational constant

Corrected

Reviewer 2 Report

The reviewed work uses two radial density models, the PREM and the AK135-F, for the estimation of the Earth’s total mass and gravity field parameters (geoidal geopotential number W0 and mean gravitational attraction). From the numerical results, it is concluded that both models agree very closely with the published Earth’s parameters. However, some minor revisions should be made, especially in the Equations, in order to improve the manuscript, as follows:

  • Line 114: The density parameters up to the third order in the Table 1, should be referred in this paragraph.
  • Line 154: The symbol “m” should be replaced with “R”.
  • Line 157: Eq. (3) is wrong and may be replaced with the correct one (e.g. symbol “V” is omitted, the parameter G is omitted from the first two terms of this equation).
  • Line 158: The symbol “G” is omitted.
  • Line 161: The sign in the second term can be replaced with “-”.
  • Line 163: The first-term on the right-hand side of Eq. (3) is different from Eq. (5).
  • Line 169: The first-term on the right-hand side of Eq. (4) is different from Eq. (7).
  • Line 199: The mass of the Earth can be written correctly as 10^24.
  • Line 237: The title “Roche’s model” can also be referred in the relative equation.

Author Response

Annotated respond

We thank Editors for their constructive comments. We addressed them below.

The reviewed work uses two radial density models, the PREM and the AK135-F, for the estimation of the Earth’s total mass and gravity field parameters (geoidal geopotential number W0 and mean gravitational attraction). From the numerical results, it is concluded that both models agree very closely with the published Earth’s parameters. However, some minor revisions should be made, especially in the Equations, in order to improve the manuscript, as follows:

Line 114: The density parameters up to the third order in the Table 1, should be referred in this paragraph.

Here is the reference:

Dziewonski AM, Anderson DL (1981) Preliminary reference Earth model. Phys Earth Planet Inter 25 (4): 297-356

Line 154: The symbol “m” should be replaced with “R”.

Corrected

Line 157: Eq. (3) is wrong and may be replaced with the correct one (e.g. symbol “V” is omitted, the parameter G is omitted from the first two terms of this equation).

Corrected

Line 158: The symbol “G” is omitted.

Added

Line 161: The sign in the second term can be replaced with “-”.

Corrected

Line 163: The first-term on the right-hand side of Eq. (3) is different from Eq. (5).

We revised the Eq.(3)

Line 169: The first-term on the right-hand side of Eq. (4) is different from Eq. (7).

We revised the Eq.(4) and (7)

Line 199: The mass of the Earth can be written correctly as 10^24.

Corrected

Line 237: The title “Roche’s model” can also be referred in the relative equation.

Reference added

Reviewer 3 Report

This is a study focused on an assessment of the consistency of existing 1-D density models (i.e., PREM and AK135F) with reference values for Earth's mass and geoid. The introduction provides a detailed description of the state of the art of Earth's density models. In the description of the reference density models, the authors explicitly state that the PREM structure was developed accounting for total Earth's mass and volume and other geophysical constraints. I'm wondering then whether this study would be necessary since the Earth's mass was "taken into consideration" in the design of the PREM structure. It is not surprising that the authors found that the resulting Earth's mass is consistent with the reference value. If this interpretation of the manuscript main results is incorrect, the authors should better describe in the second paragraph the assumptions and measurements that were used to implement this 1-D density model. 

The Section on the methods should also be improved since there are some errors reported in the attached file. The approach used in this study to determine either the mass or the geoid is not well-described. The density model is 1-D, therefore, I'm wondering whether the authors considered Earth's asphericity (e.g., oblateness) of its external and internal layers, or they assumed it as a sphere. A further discussion on the results should be presented by the authors to better support the finding of this study, which seems to be a cross-check of existing density models only. 

For these reasons, my overall recommendation is to reconsider this manuscript after major revision.

Author Response

Annotated respond

We thank the Editors for their constructive comments. We addressed them below.

This is a study focused on an assessment of the consistency of existing 1-D density models (i.e., PREM and AK135F) with reference values for Earth's mass and geoid. The introduction provides a detailed description of the state of the art of Earth's density models. In the description of the reference density models, the authors explicitly state that the PREM structure was developed accounting for total Earth's mass and volume and other geophysical constraints. I'm wondering then whether this study would be necessary since the Earth's mass was "taken into consideration" in the design of the PREM structure. It is not surprising that the authors found that the resulting Earth's mass is consistent with the reference value. If this interpretation of the manuscript main results is incorrect, the authors should better describe in the second paragraph the assumptions and measurements that were used to implement this 1-D density model. 

In the forthcoming study (not in this MS), we want to use the earth observation data (e.g., satellite gravity data) to strip the disturbing gravity signal (we can use PREM and Litho1.0 model to compute) to investigate the sub-lithosphere structures. To do the forthcoming study we have to assessment the accuracy of the PREM and AK135-F radial density models.

The Section on the methods should also be improved since there are some errors reported in the attached file. The approach used in this study to determine either the mass or the geoid is not well-described. The density model is 1-D, therefore, I'm wondering whether the authors considered Earth's asphericity (e.g., oblateness) of its external and internal layers, or they assumed it as a sphere. A further discussion on the results should be presented by the authors to better support the finding of this study, which seems to be a cross-check of existing density models only. 

We thank the reviewer comments.

All the errors are corrected in the revised manuscript marked by red color.

Mass is computed by equation (2).

In this study, we do not compute the geoid but the geopotential. We do not consider the Earth's asphericity.

Yes, in this study, we assessment the accuracy of the PREM and AK135-F radial density models

Round 2

Reviewer 3 Report

I would like to thank the authors for the response letter addressing my major concerns on the manuscript. I recommend publication of the manuscript in the present form.

Sincerely,

Antonio Genova